# Influence of Halal Slaughter on the Colour, Odour, Flavour, and Textural Properties of Beef

**DOI:** 10.3390/ani15091227

**Published:** 2025-04-27

**Authors:** Said Bouzraa, Estrella I. Agüera, Carmen Avilés-Ramírez, Salud Serrano

**Affiliations:** 1Food Science and Technology Department, Faculty of Veterinary, University of Córdoba, 14071 Córdoba, Spain; v92avrac@uco.es (C.A.-R.); bt2sejis@uco.es (S.S.); 2Halal Institute Spain, EGCH S.L., 14001 Córdoba, Spain; 3Cellular Biology, Physiology, and Immunology Department, Faculty of Veterinary, University of Córdoba, 14071 Córdoba, Spain; ba1agbue@uco.es

**Keywords:** beef meat, slaughtering method, stunning, meat quality, halal

## Abstract

The objective of this study was to investigate the influence of the slaughter method on beef sensorial attributes (odour, colour, and flavour) and textural properties. Three slaughter procedures were compared, as follows: regular or conventional slaughter (with penetrative captive bolt), halal slaughter, and halal slaughter with nonpenetrative captive bolt (reversible stunning accepted by some halal religious authorities). Meat from halal slaughter with stunning showed lower scores for urine and milk odours, metallic flavour, and red (a*) and yellow (b*) indices, particularly in the *Transversus abdominis* muscle. Although current legislation authorizes religious slaughter, exceptionally, without stunning, it remains a controversial issue from the point of view of animal welfare with enormous social implications. Therefore, research is needed to develop stunning methods compatible with halal slaughter while maintaining meat quality.

## 1. Introduction

Food quality is increasingly determined not only by safety and nutritional value but also by animal welfare prior to slaughter. Studies suggest that improved animal welfare enhances meat quality and resistance to disease, thereby directly influencing food safety. Although European regulations allow religious slaughter without stunning under exceptional circumstances, the practice is subject to ongoing debate regarding its implications for animal welfare and societal perception [1,2,3].

From the scientific field, this controversy is equally complex. Schulze et al. [4] defended halal slaughter as a compatible method with animal welfare in their comparative study between different slaughter methods (without and with stunning). In the same way, Pouillaude-Bardon [5] concluded that slaughter without stunning was the most natural and least traumatizing method. However, during animal slaughter without stunning, the cutting of soft tissues (muscle, tissue, and viscera) of the neck induces the nociceptive nerve fibres to generate electric impulses, which are transmitted to the higher centre of the nervous centre for its interpretation as pain [6]. The basic requirements of halal slaughter are that the animal must be alive at the time of slaughter, that any preslaughter handling should not cause death to the animal, and that the stunning should be reversible, with animals gaining full consciousness if not slaughtered.

In our study, we compared meat quality attributes based on slaughter type considering whether there was stunning. Opponents of the stunning method maintain that it is a practice contrary to Islamic law and could lead to animal death prior to slaughter—a very controversial debate, since there are authors who have explained how the stunning facilitates the correct immobilization of the animal to carry out the cutting of the blood vessels [7], in addition to reducing the risk of occupational accidents to the operator.

This has led to a clash between Islamic jurists as to whether the modern technologies such as preslaughter stunning may be accepted as part of Halal slaughter [5]. In fact, a different kind of stunning, a reversible, nonpenetrative captive bolt, for halal slaughtering according to Malaysian halal standards [8] is accepted under strict control and check of each skull to evaluate the possible damage caused. If irreversible damage is done, the carcass is rejected; hence, it is guaranteed that the obtained meat reaches the halal state.

Meat quality is generally described in terms of aesthetic, tactile, masticatory, functional, nutritional, health, convenience, and environmental-impact attributes. These can be affected by religious slaughter practices associated with preslaughter, slaughter, and postslaughter [9,10]. These terms are related to the organoleptic or sensory meat quality [11], which is defined as the characteristics perceived by the senses at the time of purchase or consumption that influence sensory satisfaction [12]. Organoleptic quality is given by enormously variable parameters, easily modifiable, objective and measurable, intrinsic to the very nature of the flesh. The organoleptic characteristics that influence the palatability of meat are, fundamentally, the texture [13], the juiciness, the flavour, and the colour. These attributes are influenced, as already mentioned, by productive and technological factors.

Sensory analysis deals with the measurement and quantification of product characteristics that are perceived by the human senses [14]. To obtain a generalized methodology, current applications of sensory methods are aimed at establishing correlations with instrumental parameters [15], and thus, several studies have sought objectivity for panellists so that their assessments correspond to instrumental measures [16,17]. The sensory analysis of foods is carried out according to different tests depending on the purpose.

Of instrumental measures, colour is an important indicator of carcass quality, since visible changes in colour occur in muscle during its cooling that can influence its acceptability to the consumer at the time of purchase [18]. The colour is the overall impression that is observed of meat, resulting from the interaction among saturation (determined by the amount of pigment, mainly myoglobin), shade (determined by the chemical state of the pigment), and lightness (determined by the physical state of the meat). In addition to its importance in the evaluation of fresh meat, it is necessary to consider colour changes after cooking due to the heat treatment to which it is subjected during the cooking process [19], which the consumer relates to the degree of cooking and level of pathogen destruction.

The water-holding capacity (WHC) is especially important under the sensory point of view because of its association with the so-called juiciness [20], since the juiciness of the meat, as perceived by the consumer, is determined by the amount of water maintained in its structure at the time of consumption after maturation, storage, and cooking process [21]. Furthermore, tenderness is one of the most valued characteristics in sensory analysis, and it is one of the main factors in the acceptance or rejection of meats. The key factors affecting meat tenderness are connective tissue and cross-links, myofibrillar integrity, sarcomere length, protein denaturation, and intramuscular fat content [22]. Regarding tenderness, it is known that this is the meat attribute that the consumer most appreciates at the time of consumption. Tenderness allows meat to be easily cut and chewed, and is directly linked to the mechanical resistance of the consumable product, while hardness indicates persistent resistance to breaking during chewing [23].

This study aims to determine the influence of the slaughter method on the sensory attributes (odour and flavour), instrumental colour, and textural properties of beef steaks. The tested slaughter systems were slaughter with previous stunning with penetrating captive bolt gun, halal rite slaughter with reversible stunning, and halal rite slaughter without stunning.

## 2. Materials and Methods

### 2.1. Sampling and Slaughtering Conditions

The studied samples consisted of two types of muscles: the *Transversus abdominis* and *pectoralis*. The muscles were obtained from 45 carcasses of calves (the averages for live weight and carcass weight were 600 and 340 kg, respectively). The slaughtered animals were Limousine breed males aged between 12 and 14 months and came from the same farm under similar handling conditions. Detailed information about the animals was obtained from the purchase documents of the slaughterhouse (COVAP, Córdoba, Spain). The animals were slaughtered according to the meat industry protocol.

Three types of slaughter techniques were used to obtain the muscle samples:Standard or regular procedure, which involves stunning the animals mechanically using a penetrating captive bolt gun stunning (15 heads). These samples were identified as conventional (CONV).Halal ritual procedure, in which slaughtering is performed without stunning (15 heads). These samples were identified as nonstunning halal slaughter (NHS). Islam teaches that animals are to be slaughtered according to the mindful and attentive way taught by the prophet (prophetic method), using a sharp knife to quickly severe the four vessels in the neck (carotids and jugulars, trachea, and oesophagus), mentioning the name of God during the slaughtering with sincerity and conviction, such that the animal is dispatched as painlessly as possible [9].Halal ritual procedure using acceptable reversible stunning with nonpenetrating pneumatic gun (halal-acceptable stunning; HAS). In this case, the cut is performed immediately after stunning (1–2 s). A Karl Schermer Type KC (Germany), a type of nonpenetrative captive bolt, was used for reversible stunning for halal slaughtering. Following ignition of the cartridge, the propellant charge accelerates the bolt to such a strong extent that the impact plate strikes the skullcap of the animal at a speed of about 45–65 m/s [24].

Twenty-four hours after animal slaughter, the muscles were taken from the carcasses, vacuum packaged, and placed in portable refrigerators for transport at 4 °C to the laboratory. Then, samples were preserved at −24 °C until being processed. After thawing, the samples were divided into steaks (45 steaks were destined to sensory evaluation, and another 45 steaks were destined to instrumental study through different tests).

### 2.2. Sensory Analysis

This section describes in detail the procedure for the sensory evaluation of beef samples, especially regarding the preparation of the samples, the descriptive profile used, and the test execution conditions. In this type of analysis, it is important to standardize the tasting conditions to minimize the experimental error.

#### 2.2.1. Tasting Session Preparation

Prior to the sensory evaluation session, the responsible verified that the conditions of the tasting room were standardized and appropriate, recording the following checks: that the temperature of the tasting room and sample preparation room was maintained between 18 and 25 °C by thermometer, that the sample preparation room had the material necessary, and that each cabin in the room had the necessary material for the organoleptic assessment session.

#### 2.2.2. Sample Preparation

All samples were cut to a uniform thickness (2.5 cm). The thawing of samples took place during 24 h in a refrigerated room at 4 °C. When presenting the samples to the assessors, it was necessary to remove the fat from coverage of the sample and connective tissue, since these parts were not the target of the study. Unsalted breadsticks and sips of natural mineral water were used as flavour cleaning agents [17].

Samples were cooked on a clamshell-type electrical grill (Grill Comfort, Tefal, Rumilly, France) until the centre of each sample reached a temperature of 70 °C [25]. This was controlled using a Hanna Checktemp1 (HANNA instruments S.L., Gipuzcoa, Spain) introduced in the geometric centre of the sample. Cooked sections were wrapped in aluminium foil. Then, samples were cut into the necessary parts according to the number of assessors. The samples presented a heterogeneous geometry depending on the shape of the original sample, with 1 cm^2^ being the presentation base for a length of 2.5 cm [26]. Once chopped, they were wrapped in aluminium foil. Each sample was identified with a permanent marker and kept warm until analysis.

#### 2.2.3. Sensory Descriptive Profile

The attributes considered in the present study for the sensory evaluation are listed in Table 1.

The olfactory attributes were evaluated immediately at first smell in regard to their presence and intensity. The flavour attributes were evaluated as retronasal perceptions assessed during the chewing process from a piece of tempered sample. The quantification of the descriptors was carried out using a descriptive test with an unstructured hedonic scale of 10 cm in length, which consisted only of extreme points, that is, minimum and maximum appreciation of the food attribute intensity [27]. Once the samples were properly thawed, muscles were cut perpendicular to the muscle fibre orientation to produce as many 2.5 cm thick steaks as possible.

#### 2.2.4. Development of Sensory Analysis

To evaluate the samples, 8 trained assessors were selected to evaluate 9 samples per session. A randomized complete block design was used, and 3 samples per plate and round were analysed, with 3 rounds each session. The sensory evaluation was carried out in five sessions, in standardized sensory cabins [28]. This panel was set up with individuals from different ages and genders (4 men and 4 women) tasked with evaluating and analysing the sensory characteristics of the samples, often focusing on odour and flavour to gather data on the sensory qualities, understand consumer preferences, and improve meat quality.

### 2.3. Instrumental Analysis

Complementing the sensory profile, an instrumental study was carried out to obtain accurate data, since instrumental analysis is objective and relatively easy to perform.

#### 2.3.1. Instrumental Colour

The colour parameters of the samples, lightness (L*), red index (a*), and yellow index (b*), were measured using a CM2600d portable spectrocolorimeter (Minolta Co., Osaka, Japan) calibrated with a white plate (L * = 97.78, a * = 0.19, b * = 1.84) and light trap supplied by the manufacturer. After 30 min of oxygenation (blooming) at room temperature, three measurements were taken in homogeneous, nonoverlapping areas of each steak, which represented the entire surface of each sample, avoiding areas of connective tissue and intramuscular fat. Finally, the average of the three measurements was calculated.

#### 2.3.2. Water-Holding Capacity Measurement

The water-holding capacity (WHC) of the samples was assessed evaluating cooking losses. To do this, before cooking, the steaks were weighed on a scale (HGS-3000 series, Mettler, Toledo, Spain), and a Hanna Checktemp1 (HANNA instruments S.L., Gipuzcoa, Spain) thermometer with a flexible high-temperature probe was placed in the geometric centre of each steak. Then, each steak was cooked on a double-plate electric griddle with a grill heater set at 180 °C. During the process of cooking, the temperature was monitored every 30 s, the steaks were not turned over, and the opening of the top lid was minimized. Once the desired final temperature (70 °C) was reached in the centre, samples were removed from the grill, cooled at room temperature, placed in individual plastic bags of polyethylene, then stored refrigerated at 4 °C for 24 h. After this time in refrigeration, the samples were removed from the bags and tempered at room temperature (approximately 20 °C), then dried with paper filters without compression and weighed again. Cooking losses were obtained using the following formula (where initial weight corresponds to grams of meat before cooking and final weight corresponds to grams of meat after cooking):Meat cooking losses = [(Initial weight − final weight)/initial weight] × 100

#### 2.3.3. Instrumental Texture Measurement

For the analysis of the Warner–Bratzler shear force (WBSF), the samples were cooked, preserved, and tempered in the same way as in the previous procedure. Once processed, each sample was cut into ten segments of 1 × 1 × 3 cm (height, width, length). These pieces were used to measure the shear resistance using a TA-XT-2 Texturometer (Texture Analyzer^®^, Stable Micro Systems, Surrey, UK) equipped with a Warner–Bratzler blade with a 30 kg load cell and a cutting speed of 200 mm/min. The cutting distance was 3 cm (the blade had to completely cut the meat sample). The parameter recorded was the maximum resistance presented by the sample at the cut, which coincided with the maximum peak force (Figure 1) (in kg/cm^2^). The shear force was measured cutting perpendicular to the muscle fibres, avoiding connective tissue and fat parts. Then, the peak cutting force values of the ten segments in each sample were averaged.

Through this analysis, graphs such as the one shown in Figure 1 were obtained.

### 2.4. Statistical Analysis

Statistical analysis was carried out with the STATISTICA software, version 12, StatSoft, Inc., Tulsa, OK, USA (2014), evaluating the effect of the type of stunning on the sensory and instrumental quality of the meat through a comparative analysis of means (ANOVA). When significant differences were found, a multiple comparative analysis of means was carried out through a Tukey’s HSD test. Differences between samples were considered significant when the probability of rejecting the null hypothesis was equal to or less than 0.05. A multivariate analysis was performed with the PCA command of the XLSTAT 2020 software (Addinsoft Inc., New York, NY, USA). We carried out a principal component analysis (PCA) using a Pearson correlation matrix on the mean values for descriptive measures of the sensory analysis and instrumental measurements.

## 3. Results and Discussion

To verify the existence of differences in the sensory analysis due to the type of stunning, an analysis was carried out comparing each type of slaughter. The average values are shown in Table 2.

It was observed that the assessors perceived significant differences (*p* < 0.05) in the taste of milk between *Transversus abdominis*—HAS and the other two treatments and in the taste of urine between the *Transversus abdominis*—NHS and *Transversus abdominis*-HAS, as well as a trend toward significance (*p* < 0.1) for other odours between *Transversus abdominis*—HAS and the other two treatments. For liver and blood odours there were no significant differences between the three types of slaughter. Regarding the sensory evaluation of the flavour, there was only a significant difference for the metallic attribute between the *Transversus abdominis*—NHS and *Transversus abdominis*—CONV treatments.

The liver odour and flavour were the two attributes that were appreciated with higher intensity, with scores between 2.3 and 3.3 and between 2.0 and 3.1, respectively. The values for liver flavour were higher than those observed in another study [29], although it is true that the commercial cut used was loin.

Very few references were found on sensory evaluation of individual muscles. In this sense, Carmack et al. perceived the intensity of *pectoralis* muscle beef as low (6.7 on a 10-point scale) [30]. Our results for liver flavour in the *pectoralis* muscle were lower than those (2.5–3.1), with the liver flavour being associated to the beef flavour.

Figure 2 visually summarizes odour and flavour profiles, highlighting significant differences in milk and urine odours, and metallic flavour, particularly in *Transversus abdominis* muscle. The variable “other attributes” represents other odours/flavours that were detected by the panellists, with odours such as toast, grass, grease, caramel, and manure and flavours such as toasted, milk, slight dairy flavour, butter, no flavour, and exquisite aftertaste.

The consumer’s choice at the point of purchase depends largely on the external appearance and, consequently, colour. Furthermore, the instrumental texture is related to tenderness, which is the first parameter that the consumer values when evaluating the postconsumption satisfaction of beef, hence the importance of measuring these parameters.

Colour was correlated with the intensity coordinate L, red–green a*, and yellow–blue b*, where luminosity and colour had greater acceptance, constituting direct and efficient commercial acceptance of fresh meat [31].

As can be seen in Table 3, the slaughter method did not have a significant effect (*p* > 0.05) on the instrumental parameters of the two cuts of beef tested. Still, regarding the red index, the differences observed between the different slaughter systems showed a trend toward significance (*p* < 0.1)—animals slaughtered by halal with reversible stunning presented less red meat than those slaughtered by the halal method without stunning, with animals slaughtered by the conventional method showing a red index intermediate between the other two groups. Although there are studies that have confirmed that animals slaughtered by the halal rite without stunning show more efficient bleeding, which results in carcasses with less blood (less haemoglobin, with inferior L* and a* indices), there is a great deal of controversy around this fact, which is why it is complex and risky to draw conclusions from our results [5]. For the yellow index, animals slaughtered by halal with reversible stunning presented a lower value than those slaughtered by the halal method without stunning or by the conventional method.

Cooking losses, due to thermal denaturation of the meat proteins [20], constitute an important problem for the food industry. A study by Yang et al. [32] showed that denaturation of proteins takes place in three steps: denaturation of myosin (40–60 °C), denaturation of sarcoplasmic protein and collagen (60–70 °C), and denaturation of actin (approximately 80 °C). Heat causes contraction [33] and pronounced loss of fibres, leading to substantial water loss (15 to 35%). Table 3 shows the average values of cooking losses, without significant differences attributable to the type of slaughter. Similar cooking losses were determined by Garmin et al. [34] for the *Transversus abdominis*. Önenç and Kaya [35] also did not show significant cooking losses between treatments at 24 h and 4 days postmortem, although they showed significant differences at 7 and 14 days postmortem (*p* < 0.05).

There were no significant differences in texture among the three slaughter methods. The shear strength values in this study were higher than those published by Belew et al. [36] for the diaphragm muscles (20.3 kg/cm^2^) and *Transversus abdominis* (44.3 kg/cm^2^), although it is true that the samples from that study were ripened for 14 days, unlike the samples in our study, which were processed 24 h postmortem.

According to Valin and Ouali [37], the hardness of the meat is determined by the characteristics of connective tissue and myofibrillar proteins and cytoskeleton. Variations in the texture of the meat can be related to antemortem aspects such as animal species, breed, sex, age, animal nutrition, or even slaughter weight or susceptibility to stress prior to slaughter of animals. Furthermore, as detailed by Koohmaraie et al. [38] and Descalzo et al. [39], among other authors, the texture is strongly related to postmortem physical changes in skeletal muscle structures (myofibrillar structure), the enzymatic activity during the maturation of the meat, with the amount of connective tissue, changes in the length of the sarcomeres, and to the fat content in the inter- and intramuscular spaces.

A principal component analysis (PCA) was performed on sensory attributes and instrumental measurements. Beef samples were differentiated both by slaughter method and by muscle (Figure 3). The two principal components accounting for 57.31% of the total variance described the variation in the parameters studied of the beef samples. Samples of *Transversus abdominis* muscle tended to receive positive scores on the PC1 axis, which explained 34.25% of the variance. Regarding the slaughter method, HAS samples were separated from NHS and CONV on the PC2 axis, with the former samples showing positive scores for both muscles. These findings align with those of other authors who evaluated the bleeding efficiency in cattle slaughtered by captive bolt stunning and neck cut vs. traditional halal slaughter without stunning and reported comparable blood loss variables, packed cell volume, and meat quality between both groups [40].

Meat quality, which is ultimately set by the consumer, is determined by several factors and is directly related to the survival and development of human beings. It is the most critical aspect for consumers to consider when purchasing meat. Some consumers are even willing to pay a higher price to guarantee the quality of meat [41]. Among the sensory attributes, colour, tenderness, and flavour, in that order, are the traits that most influence the acceptability of meat.

A study carried out on the characteristics of meat consumption by European consumers [42] concluded that tenderness is the critical factor in the appreciation of the quality of beef and that the flavour can become important only when a certain threshold of tenderness has been reached. Furthermore, it has been demonstrated that consumers are willing to pay a premium for those meats that have guaranteed tenderness [43].

## 4. Conclusions

This study revealed that, overall, the slaughter method had limited influence on the sensory and textural properties of beef. Our results revealed no significant differences for sensory evaluation (odour and flavour) and textural properties between halal with and without stunning and conventional slaughters, excepting urine and milk odours, metallic flavour, and a trend toward significance for red and yellow indices only for the *Transversus abdominis* muscle. For this muscle, beef samples from halal-acceptable stunning (HAS) demonstrated lower scores for undesirable odours and metallic flavour and tended to show lower red and yellow indices than samples from nonstunning halal slaughter (NHS) and conventional slaughter (CONV). The type of slaughter did not affect liver or blood odours/flavours, nor the water-holding capacity measured from the cooking losses.

Universally accepted rules and practices for religious slaughter are still under debate, and the certification and labelling of meat products remain as issues to be addressed. Therefore, measures are being taken to minimize animal welfare concerns by improving slaughter practices through increased training and the implementation of new regulations. For this purpose, more and deeper studies are required not only on animal welfare at the time of slaughtering but on meat quality in general. Greater awareness and scientific validation of these practices can help support informed decision-making by regulatory bodies, consumers, and meat producers.

## Figures and Tables

**Figure 1 animals-15-01227-f001:**
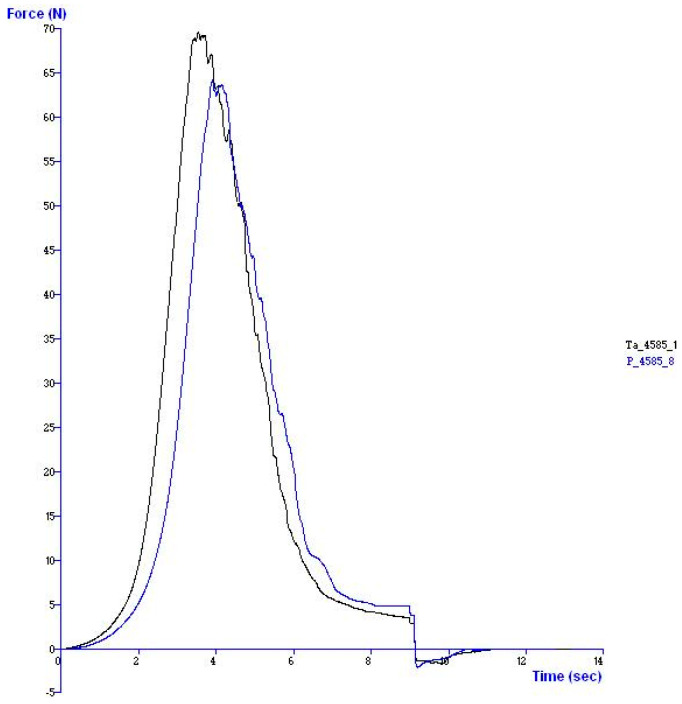
Graphic presentation of Warner–Bratzler test results, showing the maximum peak force value of two different samples registered in a test.

**Figure 2 animals-15-01227-f002:**
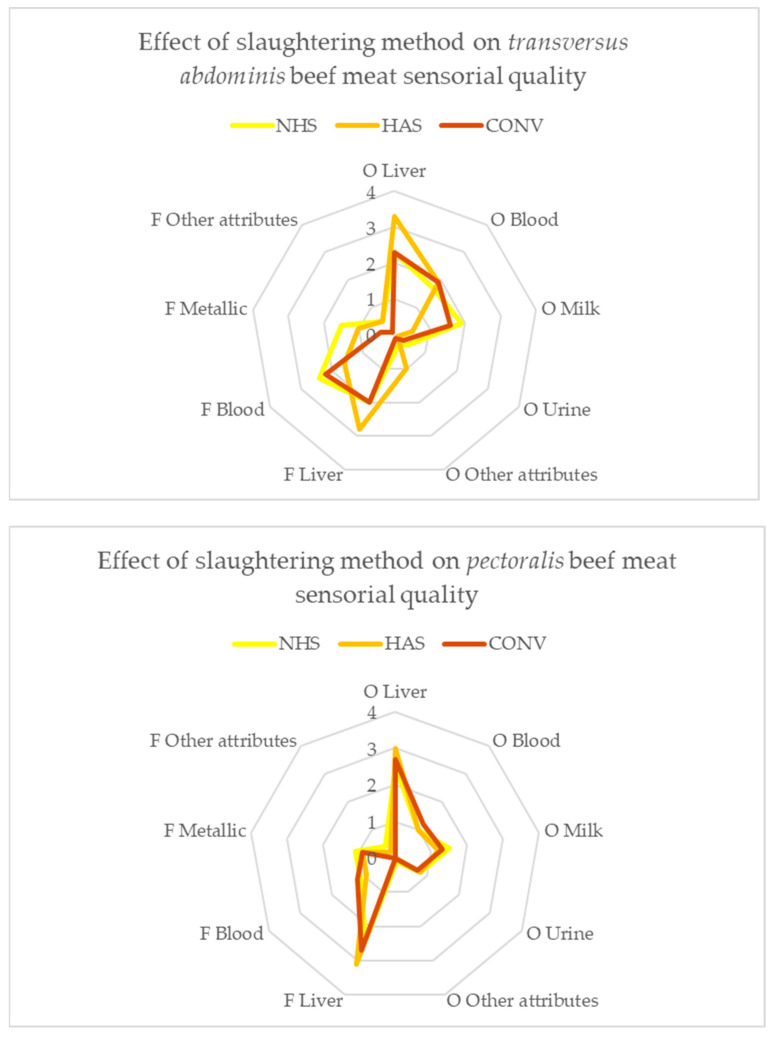
Graphic representation of the odour and flavour profiles (Table 2).

**Figure 3 animals-15-01227-f003:**
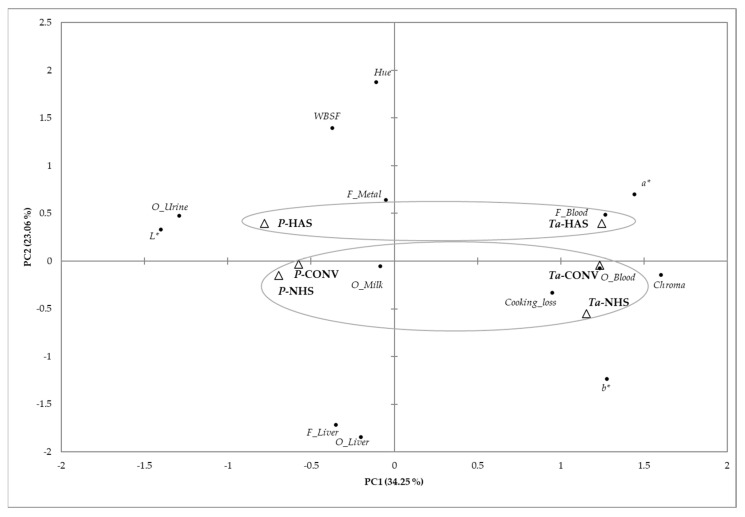
Principal component analysis plot representing the differentiation of beef samples (*Transversus abdominis*, Ta, and *pectoralis*, P, muscles) of animals slaughtered by 3 different methods (halal acceptable stunning slaughter, HAS; nonstunning halal slaughter, NHS; and conventional, CONV) based on the main sensory attributes and instrumental colour and texture.

**Table 1 animals-15-01227-t001:** Odour and flavour attributes for sensory evaluation of bovine meat.

Odour Attributes	Description
Liver	odour like cooked liver
Blood	odour of warm blood from a recently slaughtered animal
Milk	clear perception of the characteristic odour of lactic acid, meat from young animals fed mainly on breast milk, and even the odour of sugar or candy
Urine	pungent, slightly sour, and ammoniacal odour reminiscent of urine
Others	if the assessor clearly perceives another odour/s, it should be described, and its intensity should be assessed
Flavour attributes	Description
Liver	cooked liver
Blood	fresh blood flavour, like that produced by aqueous solutions diluted with plasma powder
Metallic	flavour associated with the presence of iron in the sample, like that of ionic iron

**Table 2 animals-15-01227-t002:** Effect of the slaughter method (means ± SD) on the sensory evaluation of two different cuts of beef (*Transversus abdominis* and *Pectoralis* muscles). Scores were obtained from a 10 cm non structured scale.

	** *Transversus abdominis* ** **—NHS**	** *Transversus abdominis* ** **—HAS**	** *Transversus abdominis* ** **—CONV**	** *p* ** **-Value**
Odour				
Liver	2.3 ± 0.8	3.3 ± 0.7	2.3 ± 1.2	ns
Blood	1.7 ± 0.5	1.9 ± 1.5	1.9 ± 1.0	ns
Milk	1.9 ± 1.1 ^a^	0.5 ± 0.4 ^b^	1.6 ± 0.4 ^a^	*
Urine	0.5 ± 0.1 ^a^	0.1 ± 0.1 ^b^	0.3 ± 0.3 ^ab^	*
Other	0.3 ± 0.3 ^b^	1.0 ± 1.1 ^a^	0.1 ± 0.1 ^b^	t
Flavour				
Liver	2.0 ± 0.8	2.8 ± 1.1	2.0 ± 1.2	ns
Blood	2.4 ± 0.3	1.6 ± 1.0	2.2 ± 1.2	ns
Metallic	1.5 ± 0.5 ^a^	1.0 ± 0.6 ^ab^	0.4 ± 0.	*
Other	0.5 ± 0.4	0.5 ± 0.6	0.1 ± 0.2	ns
	** *Pectoralis* ** **—NHS**	** *Pectoralis* ** **—HAS**	** *Pectoralis* ** **—CONV**	
Odour				
Liver	2.6 ± 1.2	3.0 ± 1.0	2.7 ± 1.1	ns
Blood	0.9 ± 0.3	1.0 ± 0.3	1.2 ± 0.4	ns
Milk	1.4 ± 0.7	1.2 ± 0.6	1.3 ± 0.6	ns
Urine	0.8 ± 0.2	0.8 ± 0.3	0.7 ± 0.2	ns
Other	0.1 ± 0.3	0.0 ± 0.1	0.0 ± 0.1	ns
Flavour				
Liver	2.5 ± 1.2	3.1 ± 1.0	2.7 ± 1.2	ns
Blood	1.1 ± 0.5	0.9 ± 0.3	1.2 ± 0.5	ns
Metallic	1.1 ± 0.6	0.9 ± 0.5	0.9 ± 0.3	ns
Other	0.4 ± 0.5 ^a^	0.2 ± 0.3 ^a^	0.0 ± 0.0 ^b^	t

NHS: nonstunning halal slaughter; HAS: halal-acceptable stunning; CONV: conventional slaughter; t (trend): *p*-value < 0.1; *: *p*-value < 0.05; ns: not significant; ^a,b^: values followed by the same letter within the same row are not significantly different (*p* > 0.05) according to Tukey’s HSD test.

**Table 3 animals-15-01227-t003:** Effect of the slaughter method (means ± SD) on the instrumental parameters of two different cuts of beef (*Transversus abdominis* and *Pectoralis* muscles).

	** *Transversus abdominis* ** **—NHS**	** *Transversus abdominis* ** **—HAS**	** *Transversus abdominis* ** **—CONV**	** *p* ** **-Value**
L	27.6 ± 5.6	22.6 ± 4.3	29.0 ± 5.1	ns
a	20.3 ± 1.2 ^a^	17.4 ± 2.9 ^b^	18.3 ± 2.5 ^ab^	t
b	16.2 ± 2.4 ^a^	14.4 ± 0.4 ^b^	16.5 ± 2.5 ^a^	t
Chroma	26.0 ± 2.0 ^a^	22.6 ± 2.1 ^b^	24.7 ± 3.5 ^ab^	t
Hue	51.6 ± 4.2	50.1 ± 5.3	48.0 ± 1.4	ns
WBSF (Kg/cm^2^)	80.9 ± 12.3	66.0 ± 17.1	79.3 ± 19.0	ns
Cooking loss %	25.2 ± 6.1	28.5 ± 3.6	27.5 ± 3.4	ns
	** *Pectoralis* ** **—NHS**	** *Pectoralis* ** **—HAS**	** *Pectoralis* ** **—CONV**	
L	35.7 ± 3.3	36.1 ± 2.1	36.7 ± 3.1	ns
a	11.9 ± 3.5	12.8 ± 2.7	12.4 ± 3.3	ns
b	7.7 ± 3.7	9.9 ± 2.9	9.7 ± 2.8	ns
Chroma	14.7 ± 2.7	16.6 ± 1.0	16.0 ± 2.8	ns
Hue	56.9 ± 16.6	52.0 ± 13.4	51.8 ± 11.6	ns
WBSF (Kg/cm^2^)	92.5 ± 16.2	83.8 ± 19.4	80.1 ± 17.8	ns
Cooking loss %	22.9 ± 2.6	20.9 ± 4.1	21.4 ± 4.9	ns

NHS: nonstunning halal slaughter; HAS: halal-acceptable stunning; CONV: conventional slaughter; t (trend): *p*-value < 0.1; ns: not significant; ^a,b^: values followed by the same letter within the same row are not significantly different (*p* > 0.05) according to Tukey’s HSD test.

## Data Availability

The original contributions presented in this study are included in the article. Further inquiries can be directed to the corresponding author.

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
