# Peer review of "Influence of Halal Slaughter on the Colour, Odour, Flavour, and Textural Properties of Beef"

_animals, 2025, doi:10.3390/ani15091227_

Round 1
Reviewer 1 Report
Comments and Suggestions for Authors
The manuscript Influence of halal slaughter on the colour, odour, flavour and textural properties of beef analyses a topic that fits within the scope of the Journal. The topic is relevant and the centre of social discussion. However, the discussion of data has many mistakes that should have been checked prior to submission, and English has to be revised. Differences between muscles could have been analysed (not only between treatments). Finally, tenderness is missing in the sensory study.
- Affiliations should be ordered by order of mentioning. Therefore, Halal Institute should be 2, and Cellular biology should be 3.
- L13, 93. Delete cattle.
- L44-46. This sentence is repetitive with the previous one (41-43). Delete one of the two.
- The introduction should discuss different slaughter methods and how they can affect the animal and, therefore, the meat. Very few lines have focused on this aspect, most of the introduction is a general explanation of meat quality parameters.
- 1 section. An Ethics Committee approval should be indicated in this type of experiments. The name of the abattoir should be indicated. The characteristics of the animals should be indicated per treatment (at least, gender and weight, also age if possible)
- Section 2.2.3 should be before section 2.2.2.
- the average live weight was 300 kg
- ..until processed. Were the samples vacuum packaged? Indicate.
- L133-140. This can be removed; it does not add useful information about the methodology applied.
- Indicate the temperature of cooking.
- Section 2.2.4. Which design was used? Balance, unbalance, complete blocks, incomplete blocks, samples per plate…
- In fresh or defrosted meat? Indicate.
- L245-246. This is not true. Only 2 out of five attributes are significant, one out of five with a trend.
- L247-248. Tansversus HAS is 3.3 and the effect is not significant, therefore, this sentence is incorrect, liver odour is not higher.
- If the effect is not significant, there is not higher intensity of blood odour in the meat in any treatment. This discussion is incorrect.
- CONV showed similar urine odour than HAS and NHS.
- L253-254. This is not true, there were differences in metallic flavor in transversus and a trend in other in pectoralis.
- L339-359. This is the type of discussion to be set in the introduction. It is not directly related to the variables assessed in this manuscript.
- Tables 2-3. Change Values followed by the same letter 243 within the same column are not significantly different by Values followed by the same letter 243 within the same row are not significantly different
- The scale should appear in each table.
- Units have to be added to WBSF and cooking loss.
English must be revised.
Reviewer 2 Report
Comments and Suggestions for Authors
The subject of the paper is quite interesting. However the presentation is not very clear. There are some problems related to the language (please see "Coments on the Quality of English Language), but ther are also some other serious concerns. For instance:
LINES 160-161 – To clarify: the samples were obtained thawing muscles that had been at -24ºC (from section 2.1) for 24 hours? This sentence should be rewritten.
LINES 200-205 – This should be better explained (here it isn´t just a matter of being rewritten).
LINES 241-242 – This should be more complete. Maybe something like “Table 2. Effect of the slaughter method (means ± SD) on the sensory evaluation of two different cuts of beef (Transversus abdominis and Pectoralis muscles).”
LINES 243-244 – This should be more complete. Maybe something like “NHS: non-stunning halal slaughter; HAS: halal acceptable stunning; CONV: conventional slaughter; t (trend): p-value < 0.1; *: p-value < 0.05; ns: not significant; a-b: Values followed by the same letter within the same column are not significantly different (p > 0.05) according to Tukey HSD test.”
LINES 245-247 – To say “It was observed that the assessors perceived significant differences in the case of odour or tending towards significance for each of its attributes (p<0.1) for transversus abdominis muscle.” seems too simplistic and not true, since for liver and blood odour, there were no significant differences, although liver odour tended indeed to be higher for HAS.
LINES 247-254 – Some of the results here mentioned, concerning the transversus abdominis muscle, don´t seem to be in agreement with the following results shown in Table 2:
- The results for blood showed no significant differences, being similar for HAS and CONV (1.9), and higher than for NHS (1.7).
- The results for milk were significantly higher for NHS and CONV (1.9 and 1.6, respectively).
- The results for urine were significantly higher for NHS (0.5) than for HAS (0.1), but for CONV (0.3) there was no significant difference comparing to NHS and HAS.
- Considering sensory evaluation, the results for metallic flavour were significant.
LINES 255-258 – What do the authors mean by "the whole of the study popullation"? For liver odour the lower value was 2.3 (concerning transversus abdominis-NHS and transversus abdominis-CONV samples) and the higher value was 3.3 (concerning Transversus abdominis-HAS samples), respectively lower and higher than the 2.5-3.1 interval here mentioned. This interval does not even apply to the three slaughter treatments for liver odour for pectoralis samples, which was 2.6-3.0... Which studies are authors referring to? If it is just [28] is just one study. Besides, the results from [28] were indeed obtained with loin samples, but concerned flavour and not odour.
LINES 259-262 – How does this compare to the present results? Since the authors mention this, they should relate it to their own results.
LINES 363-365 – How can these conclusions be taken if the results did not show any significant differences for liver odour (despite a trend for higher liver odour concerning HAS, still with p > 0.1) and L* (here there wasn't even any trend for lower results concerniung HAS for pectoralis), and for a* and b* the results only showed a trend (p > 0.1) for lower valus concerning HAS transversus abdominis.
Comments on the Quality of English LanguageA very extensive revision is needed. Just some comments/suggestions:
LINES 41-46 – These two sentences should be joined and rewritten in an integrated and logical way.
LINES 64-67 – This should be rewritten. Maybe something like: “To obtain a generalized methodology, current applications of sensory methods are aimed at establishing correlations with instrumental parameters [11] and thus several studies seek the objectivity of the panellists so that their assessments correspond to instrumental measures [12,13].”.
LINES 69-71 – This should be rewritten. Maybe something like: “Of the different instrumental measures, colour is an important indicator of carcass quality, since visible changes in colour occur in the muscle during its cooling and influences the acceptability of the consumer at the time of making their purchase [14].”.
LINES 71-78 – This should be rewritten. Maybe something like: “The colour is the overall impression that we perceive of meat when observing it, resulting from the interaction between saturation (determined by the amount of pigment, mainly myoglobin), shade (determined by the chemical state of the pigment) and lightness (determined by the physical state of the meat). In addition to its importance in the evaluation of fresh meat, it is necessary to consider colour changes after cooking, due to the heat treatment to which it is subjected during the cooking process [15], which the consumer relates to the degree of cooking and level of pathogen destruction.”.
LINES 79-80 – This should be rewritten. Maybe something like: “(…) under the sensory point of view, due to (…)”.
LINE 84 – This should be rewritten. Maybe something like: “(…) rejection of meats (…)”.
LINE 87 – This should be rewritten. Maybe something like: “(…) known that this is the meat (…)”.
LINES 88-89 – This should be rewritten. Maybe something like: “(…) to be cut and chewed (…)”.
LINES 93-94 – This should be rewritten. Maybe something like: “(…) properties of beef steaks. The tested (…)”.
LINES 101-102 – This should be rewritten. Maybe something like: “(…) calves with the average live weight of 300 kg and younger than 12 months.”.
LINES 122-124 – This should be rewritten. Maybe something like: “Twenty four hours after animal slaughter, the muscles were taken from the carcasses and placed in portable refrigerators for transport at 4ºC to the laboratory and then preserved at -24ºC until being processed.”
LINES 124-125 – This should be rewritten. Maybe something like: “(…) divided into steaks (…)”.
LINES 128-129 – This should be rewritten. Maybe something like: “(…) for the sensory evaluation of beef samples (…)”.
LINES 148-199 – This should be rewritten. Maybe something like: “The attributes considered in the present study for the sensory evaluation are listed in Table 1.”.
LINES 163-164 – This should be rewritten. Maybe something like: “Unsalted breadsticks and sips of natural mineral water were used as flavour cleaning agents.”.
LINES 165-168 – This should be rewritten. Maybe something like: “Samples were cooked on a clam-shell-type electrical grill (Grill Comfort, Tefal, France) until the center of the sample reached a temperature of 70ºC [26]. This was controlled using a Hanna Checktemp1 (HANNA instruments) introduced in the geometric center of the sample.”.
LINES 177-180 – This should be rewritten. Maybe something like: “This panel was set up with individuals from different age and gender (4 men and 4 women) tasked on evaluating and analyzing the sensory characteristics of the samples.”
LINES 185-191 – This should be rewritten. Maybe something like: “The colour parameters of the samples, lightness (L*), red index (a*) and yellow index (b*), were measured using a CM2600d portable Spectro-colorimeter (Minolta Co., Osaka, Japan) calibrated with a white plate (L * = 97.78, a * = 0.19, b * = 1.84) and light trap supplied by the manufacturer. After 30 minutes of oxygenation (blooming) at room temperature, three measurements were taken in homogeneous, non-overlapping areas of each steak, which represent the entire surface of each sample, avoiding areas of connective tissue and intramuscular fat.”.
Round 2
Reviewer 1 Report
Comments and Suggestions for Authors
Authors have amended the manuscript after previous comments.
Author Response
Thank you. Manuscript have been amended after previous comments.
Reviewer 2 Report
Comments and Suggestions for Authors
LINES 11-13 – Beef is the meat from cattle. Maybe this first sentence could be replaced by something like: The objective of this study was to investigate the influence of the slaughter method on beef sensorial attributes (odour, colour and flavour) and textural properties.
LINE 50 – Please change “pain[6]” to “pain [6]”
LINE 139 – Maybe change “until processed” to “until being processed”
LINE 146 – Maybe change “reduce significantly” to “minimize”
LINE 149 – Maybe change “standardized, appropriate and recording the following checks” to “standardized and appropriate, recording the following checks”
LINES 155-156 – The sentence “The thawing of the samples was carried out under refrigeration for 24 hours.” remains a bit confusing - as it is, it seems that thawing took place during 24 hours in a refrigerated room... Is this what the authors want to say? Or do they want to say that the samples were kept under refrigeration for 24 hours before thawing? In that case, at what temperature?
LINE 172 – Table 1 should be clearly separated from the following text.
LINES 202-203 – Maybe change “Cooking losses was the method used to assess water holding capacity (WHC) in this study. To do this, before cooking the steaks were weighed on a scale (HGS-3000 series). After that, a thermometer with flexible high temperature probe type Hanna Checktemp1 (HANNA instruments) in the geometric center pattern of each steak and cooked on the double plate electric griddle with grill heater set at 180°C. The temperature was monitored every 30 seconds, and during the process of cooking they did not turn over and the opening of the top lid was minimized.” to “The water holding capacity (WHC) of the samples was assessed evaluating cooking losses. To do this, before cooking, the steaks were weighed on a scale (HGS-3000 series) and a thermometer with flexible high temperature probe type Hanna Checktemp1 (HANNA instruments) was placed in the geometric center of each steak. Then, each steak was cooked on a double plate electric griddle with a grill heater set at 180°C. During the process of cooking the temperature was monitored every 30 seconds, the steaks weren't turned over and the opening of the top lid was minimized. ”
LINES 213-215 – Maybe change to something like:
Cooking losses were obtained using the following formula (where initial weight corresponds to grams of meat before cooking and final weight corresponds to grams of meat after cooking):
Meat cooking losses = [(Initial weight – final weight) / initial weight] *100 (1)
LINE 225 – Please change “(in kg/cm2)” to “(in kg/cm2)”
LINES 225-228 – Maybe change “The shear force was measured cutting perpendicular to the muscle fibers, and the cut should be transversal to the fiber, avoiding connective tissue and fat parts, and finally averaged the peak cutting force values of the ten segments, in each of the samples.” to something like “The shear force was measured cutting perpendicular to the muscle fibers, avoiding connective tissue and fat parts. Then, the peak cutting force values of the ten segments in each sample were averaged.”
LINES 229-231 – Maybe change “Through this analysis, graphs such as the one shown in Figure 1 were obtained, where the graphic representation of the maximum peak force value of two different samples registered in a test is observed.” to something like “Through this analysis, graphs such as the one shown in Figure 1 were obtained.”
LINE 233 – Maybe change “Figure 1. Graphic presentation for Warner-Bratzler test parameters.” to something like “Graphic presentation for Warner-Bratzler test parameters, showing the maximum peak force value of two different samples registered in a test.”
LINES 237-239 – Maybe change “(…) a comparison analysis of means (ANOVA) was carried out. When found significant differences, a multiple comparison analysis of means from Tukey's HSD test.” .” to something like “(…) through a comparative analysis of means (ANOVA). When significant differences were found, a multiple comparative analysis of means aws carried out, through a Tukey's HSD test.”
LINES 253-254 – Maybe change to something like:
Table 2. Effect of the slaughter method (means ± SD) on the sensory evaluation of two different cuts of beef (Transversus abdominis and Pectoralis muscles). Sscores have been obtained from a 10 cm non structured scale.
LINES 259-260 – Maybe change “It was observed that the assessors perceived significant differences in the case of milk and urine odours as well as trend toward significance for other odours (p<0.1)” to something like “It was observed that the assessors perceived significant differences (p < 0.05) in the case of milk, between Transversus abdominis-HAS and the other two treatments and in the case of urine, between the Transversus abdominis-NHS and Transversus abdominis-HAS, as well as a trend toward significance (p<0.1) for other odours between Transversus abdominis-HAS and the other two treatments.”
LINES 262-263 – Maybe change “Regarding the sensory evaluation of the flavour, only the metallic attribute was significant for transversus abdominis muscle.” to something like “Regarding the sensory evaluation of the flavour, there was only a significant difference for the metallic attribute between the Transversus abdominis-NHS and Transversus abdominis-CONV treatments.”
LINE 264 – Please change “flavor” to “flavour”
LINES 291-292 – Maybe change to something like:
Table 3. Effect of the slaughter method (means ± SD) on the instrumental parameters of two different cuts of beef (Transversus abdominis and Pectoralis muscles).
LINES 293-294 – Maybe change to something like:
NHS: non-stunning halal slaughter; HAS: halal acceptable stunning; CONV: conventional slaughter; t (trend): p-value < 0.1; *: p-value < 0.05; ns: not significant; a-b: Values followed by the same letter within the same row are not significantly different (p > 0.05) according to Tukey HSD test.
LINES 295-300 – Maybe change “As can be seen in Table 3, regarding the instrumental colour indices of meat, the differences observed between the different slaughter systems showed a trend toward significance (p-value < 0.1). Animals slaughtered by halal with reversible stunning presented less red meat than those slaughtered by halal method without stunning, showing animals slaughtered by the conventional method a red index intermediate between the other two groups.” to something like “As can be seen in Table 3, the slaughter method didn't have a significant effect (p > 0.05) on the instrumental parameters of the two cuts of beef tested. Still, regarding the red index, the differences observed between the different slaughter systems showed a trend toward significance (p < 0.1) - animals slaughtered by halal with reversible stunning presented less red meat than those slaughtered by halal method without stunning, showing animals slaughtered by the conventional method a red index intermediate between the other two groups.”
LINES 302 – Maybe change “with L* and a* indices inferior” to something like “with inferior L* and a* indices”
LINES 303-304 – Maybe change “It is observed significant differences between slaughter methods for the yellow index.” to something like “For the yellow index, animals slaughtered by halal with reversible stunning presented a lower value than those slaughtered by halal method without stunning or by the conventional method.”
LINES 305-306 – Maybe change “Cooking losses, because of thermal denaturation of the meat proteins [20] constitutes 305 an important problem for the food industry.” to something like “Cooking losses, due to thermal denaturation of the meat proteins [20], constitute an important problem for the food industry.”
LINES 310-313 – Maybe change “In Table 3 they are shown the average values of cooking losses without being observed significant differences attributable to the type of slaughter like those of Önenç and Kaya [34]. Similar cooking loss was determined by Garmin et al. [35] for transversus abdominis.” to something like “Table 3 shows the average values of cooking losses, without significant differences attributable to the type of slaughter. Similar cooking loss was determined by Garmin et al. [35] for transversus abdominis. Önenç and Kaya [34] also didn't show significant cooking losses between treatments at 24 h and 4 days post mortem, although they showed significant differences at 7 and 14 days post mortem (P<0.05).”
LINES 321-322 – Maybe change “as the animal species, the breed,” to something like “as animal species, breed,”
LINES 324 – Please change “Descalzo et al. [39]among other authors” to something like “Descalzo et al. [39], among other authors,”
LINES 325 – Please change “muscle structures skeletal” to something like “skeletal muscle structures”
LINES 330-332 – Maybe change to something like:
Figure 3. Principal component analysis plot representing the differentiation of beef samples (transversus abdominis, Ta, and pectoralis, P, muscles) of animals slaughtered by 3 different methods (halal acceptable stunning slaughter, HAS,:non-stunning halal slaughter; NHS, and conventional, CONV), based on the main sensory attributes and instrumental colour and texture.
LINES 338-339 – Maybe change “slaughtered by Halal with reversible stunning samples (HAS)” to something like “HAS samples”
LINE 345 – Maybe change “The quality of the meat, which is ultimately set by the consumer” to something like “Meat quality, which is ultimately set by the consumer”
LINE 346 – The authors say “(…) perhaps the most important of all is colour [41].” The fact is that Wu et al. [41] never say this!...
LINES 346-347 – Maybe change “Tenderness and flavour, in that order, are what, after colour, most influences the acceptability of meat.” to something like “After colour, tenderness and flavour, in that order, are the traits that most influence the acceptability of meat.”.
LINES 348-350 – The authors say “A study carried out on the characteristics of meat consumption by European consumers [42] concluded that hardness is the critical factor in the appreciation of the quality of beef (…)”. This is in contradiction with what is say in line 346 and so the authors should address such apparent contradiction. In addition, Dransfield et al. [42] say that "The interrelationship of attributes could be studied more thoroughly in this than in the previous trial (Dransfield et al., 1982) when meat was obtained from commercial abattoirs in member countries and when tenderness dominated acceptability. Within a taste panel the interrelationships of flavour, tenderness and juiciness were often similar in steak and in casseroles, and acceptability was a balance of all three attributes. There was considerable variation between panels. Despite their similar cooking temperatures, acceptability at MRI and AFT was dominated by flavour, while at IPA tenderness was most important." So, what the authors say here is not correct, although a previous study of Dransfield et al. (Dransfield et al., 1982) did suggest it. Please clarify.
LINE 351 – Maybe change “it has demonstrated” to something like “it has been demonstrated”.
LINES 359-361 – Maybe change “Beef from halal slaughter with reversible stunning (HAS) demonstrated lower scores for undesirable odours and metallic flavour, and lower red and yellow indices than the rest of the groups.” to something like “For this muscle, beef samples from halal acceptable stunning (HAS) demonstrated lower scores for undesirable odours and metallic flavour, and tended to show lower red and yellow indices than samples from non-stunning halal slaughter (NHS) and conventional slaughter (NHS).”.
LINES 361-364 – The sentence “On the contrary, the meat resulting from slaughter with penetrating captive bolt gun stunning (conventional method) and halal without stunning exhibited similar higher punctuations for milk and urine odours, having higher red index value.” should be eliminated - besides the fact that what is here said for the red index value isn’t exactly true (there was just a trend and, even so, only between NHS and HAS), this sentence doesn't add any important information to the information from the previous two sentences.
Comments on the Quality of English Language
